



# PARAFOG v2.0: a near real-time decision tool to support nowcasting fog formation events at local scales

Jean-François Ribaud[1], Martial Haeffelin[2], Jean-Charles Dupont[3],

5 Marc-Antoine Drouin[4], Felipe Toledo[1], and Simone Kotthaus[2]

[1]Laboratoire de Météorologie Dynamique, Ecole Polytechnique, 91128 Palaiseau, France.

[2]Institut Pierre Simon Laplace, Ecole Polytechnique, Centre National de la Recherche Scientifique, 91128 Palaiseau, France.

[3]Institut Pierre Simon Laplace, Université Versailles Saint Quentin-en-Yvelines, 78280 Guyancourt, France.

[4]Laboratoire de Météorologie Dynamique, Ecole Polytechnique, Centre National de la Recherche Scientifique, 91128 Palaiseau, France.

Correspondence to: Jean-François Ribaud (jean-francois.ribaud@lmd.ipsl.fr)





## Abstract

An improved version of the near-real time decision tool PARAFOG (PFG2) is presented to retrieve pre-fog alert levels and to discriminate between radiation (RAD) and stratus lowering (STL) fog situations. PFG2 has two distinct modules to monitor the physical processes involved in RAD and STL fog formation and is evaluated at European sites. The modules are based on innovative fuzzy logic algorithms to retrieve fog alert levels (low, moderate, high) specific to RAD/STL conditions, minutes to hours prior to fog onset. The PFG2-RAD module assesses also the thickness of the fog. Both the PFG2-RAD and PFG2-STL modules rely on the combination of visibility observations and automatic lidar and ceilometer (ALC) measurements. The overall performance of the PFG2-RAD and -STL modules is evaluated based on 9 years of measurements at the SIRTA observatory near Paris and up to two fog seasons at the Paris-Roissy, Vienna, Munich and Zurich airports. At all sites, pre-fog alert levels retrieved by PFG2 are found to be consistent with the local weather analysis. The advanced PFG2 algorithm performs with a hit rate of about 100 % for both considered fog types, and presents a false alarm ratio on the order of 10% (30%) for RAD (STL) fog situations. Finally, the first high alerts that result in a subsequent fog event are found to occur for periods of time ranging from -120 minutes to fog onset, with first high alerts occurring earlier for RAD than STL cases.



## 1 – Introduction

According to the American Meteorology Society (2020), fog occurs when visibility at the Earth's surface is lower than 1km due to the presence of suspended water droplets. The worldwide socio-economic impact resulting from this "particular type of cloud" is just as significant as from other extreme events, such as storms (Gultepe et al, 2019). Indeed, fog may be responsible for severe disruptions at airports, including delays or even

cancellations of flights (Gultepe et al., 2009), and causes frequent road accidents and may also impact rescue management. In semi-arid and arid regions again, fog can constitute a fundamental resource as it can supply water to the local population (Gandhidasan and Abualhamayel, 2007). Therefore, accurate forecasting of fog events is essential.

The classical approach for fog forecasting used by national meteorological services relies

on numerical weather prediction (NWP) models. Although the main physical processes leading to fog occurrences are well established (Gultepe et al., 2007), accurate fog formation forecasting remains challenging for NWP models. According to Steeneveld et al. (2015), models generally struggle to accurately reproduce the timing of the fog onset, its spread, its depth and its liquid water content. Statistical methods have been used to

evaluate the possibility for an NWP model to accurately predict meteorological conditions favourable to fog formation (Menut et al., 2014; Roman-Gascon et al., 2016). The difficulties of NWP fog forecasting can be explained by the fact that fog events are driven by complex land-atmosphere interactions in the atmospheric boundary layer, where vertical resolution of NWP models is still not high enough. To simulate more

detailed information, 1D high-resolution numerical models have been used to complement the classical NWP setup, which allows specific local observations to be incorporated (e.g. Bergot et al., 2005). More recently, Large-Eddy Simulations (LES) have been used to explicitly resolve small-scales processes at play within the fog cloud (Bergot et al., 2016; Mazoyer et al., 2017; Waersted et al., 2019). Still, LES modelling is

computationally very expensive, and both microphysical and chemical parametrizations are still needed.

Another approach to forecast fog events is based upon ground- and/or space-based observations. From its top perspective, satellite imagery allows to monitor fog by combining different bands with relatively good space-time resolutions. With this regard,





Cermak and Bendix (2008) developed an operational fog/low stratus daytime scheme based on Meteosat data. Cermak and Bendix (2011) extended this approach to only discriminate ground radiation fog by introducing some microphysical hypotheses, before being adapted by Egli et al. (2017) to make it suitable for several meteorological conditions encountered over Europe. Egli et al. (2018) proposed a hybrid diurnal fog

product based on the combination of satellite images and ground-based observations. In addition, Kneringer et al. (2019) and Dietz et al. (2019) developed probabilistic fog nowcasting systems to forecast different low-visibility procedures from standard meteorological measurements available at Vienna international airport for lead times of +30 min to +120 min.

While both the aforementioned satellite- and learn-based studies do not intend to track the evolution of particular physical processes driving fog formation, ground-based observations may provide valuable key information by monitoring their true values in complement to NWP models. Ground-based observations allow to accurately measure key variable at play in a fog cloud at high temporal resolution (~ every minute). For

instance, radiation fog formation results from an aerosol-particle hygroscopic growth process illustrating the vapor-to-liquid phase change before fog onset. Based on attenuated backscatter analysis, Automatic Lidar and Ceilometer (ALC) data provide key information portraying this physical process. Haeffelin et al. (2016) developed the near-real time fog analysis tool PARAFOG (hereafter referred to as PFG1), with the objective

to predict radiation fog formation based on ALC measurements together with classical meteorological observations. During the pre-fog condition (usually 1 to 3 hours before fog), PFG1 determines a reference ALC-attenuated backscatter profile based on which the rate of change of aerosol-particle hygroscopic growth can be assessed. PFG1 retrieves pre-fog alert levels with a vertical resolution of about 15 m ranging from 0 to 400 m a.g.l.

and time resolution of one minute. PARAFOG is experimentally used at Paris international airports (Roissy-Charles de Gaulle and Orly) where it provides valuable information supporting the decision making of both weather forecasters and air traffic controllers that affect scheduling of airplanes. Several years of experience with PFG1 have highlighted some limitations, such as the monitoring of stratus lowering fogs, its

capabilities to monitor the entire fog life cycle, or even its anticipation for shallow radiation fog events near the surface.




In this study, we present PARAFOG v2 (hereafter referred to as PFG2), which is an improved and extended version of PFG1 allowing to discriminate between radiation and stratus-lowering fog formation, respectively.

• Radiation fog events (RAD) refer to fog that forms during radiative cooling at the ground surface, usually at night, in presence of anticyclonic, low wind speed, and clear-sky conditions (Gultepe et al., 2007). Due to the radiative cooling, the air just above the ground is affected by a progressive hygroscopic growth of fog condensation nuclei, turning water vapor into liquid after reaching

supersaturation, whereby reducing the surface visibility.

• When radiative cooling coincides with large-scale subsidence, the cloud base height of stratus clouds can gradually decrease down to the surface, defining stratus lowering fog events (STL). Indeed, stratus cloud top radiative cooling acts to transport larger cloud droplets downwards (while strengthening the cloud top

inversion), and permits the cloud base to subside until reaching the ground at times (Dupont et al., 2012).

This article is organized as follows: Sect. 2 introduces the measurements used as input to PFG2 and measurement sites where evaluation studies are conducted. Sect. 3 presents the new methodology developed in PFG2. Section 4 presents the PFG2 results obtained at

different European sites. The quantitative assessment of PFG2 and its relative performance is presented in Sect. 5. Finally, a summary of the both main developments and results is given in Sect. 6, along with some thoughts for potential improvements.

**2 – Sites and datasets**

**a) SIRTA and European airports**

The data used in this study are mainly based on observations collected at SIRTA (Instrumented Site for Atmospheric Remote Sensing Research, Haeffelin et al. 2005), which is a French mutli-instrumental atmospheric observatory located on a plateau 20km south of Paris (48.713°N; 2.208°E; 160m a.s.l.). SIRTA aims to study both physical and


chemical processes at play in the atmosphere with a particular emphasis on clouds, aerosols, atmospheric boundary layer processes and solar energy. Today, most of the 150 state-of-the-art instruments (active & passive remote sensing, in-situ sensors) at SIRTA are set up in an area of about 1 km². The plateau is a fog-prone region where about 35 fog events occur every year (~ 10 000 minutes per fog season), making it particularly suitable

for fog studies (Haeffelin et al., 2010; Waersted et al., 2019). PFG1 field experiments took place at SIRTA from 2006 to 2014, where a synergistic suite of instruments was designed to document the entire fog life cycle in correlation with dynamical, thermodynamical, optical and microphysical properties (e.g. Elias et al., 2009; Dupont et al. 2012; Dupont et al., 2016).

Complementing the SIRTA observations, data from major European airports regularly affected by fog are considered to test the robustness of PFG2 in a range of local environments and meteorological conditions. The present study considers the following four airports: Paris Roissy-Charles de Gaulle, Zurich, Munich and Vienna (Figure 1).

**b) Instruments and fog event statistics**

Among all the instruments deployed at SIRTA, the present study makes use of a Vaisala CL31 ceilometer (generation CLE321) providing attenuated backscatter profiles at ~910 nm as well as cloud base height using the operating procedures recommended by Kotthaus et al. (2016). In addition, scatterometers (Degreane DF20/20+/320) provide horizontal

visibility at 4 m a.g.l., whereas both temperature and relative humidity observations are recorded by an automatic weather station at 2 m a.g.l. All airports considered in this study are equipped with a Vaisala CL31, automatic weather station and visibilimeters (Table 1). The present study considers 9 years of measurements at SIRTA from 2011 to 2019, and up to two fog seasons at the Paris-Roissy, Vienna, Munich and Zurich airports

between 2014 and 2017 (Table 1).

The number of fog events for each site is derived using the Tardif and Rasmussen (2007) analysis procedure. Fog events are defined by a visibility lower than 1000 m detected for at least 30 min within a 50 min period (3 of 5 blocks of 10 min). Two fog events are



merged if there are separated by less than one hour. The events retrieved by the Tardif

and Rasmussen (2007) algorithm are considered afterwards as the reference for the PFG2 analysis. In this study, we only focus on fog events that correspond to RAD or STL, which represent more than 90% of cases regardless of the sites considered. Note also that fog events retrieved following the Tardif and Rasmussen (2007) algorithm are not considered when data are missing from either the CL31 or the meteorological station measurements

as these are required input data to PFG2. The total number of fog events considered in this study is about 250 at SIRTA (2011-2019), and up to 40 at each airport site, unevenly distributed between 2014 and 2017 (Table 1).

### 3 – PARAFOG v2.0

**a) Overview**

PFG2 has been designed to operate with relatively standard instruments, which are commonly found at national meteorological service sites, airports, and/or research observatories. The rationale for this approach is to develop a fog decision tool widely and easily applicable that enables to track the evolution of both physical and key parameters

in near-real time for fog formation. In the same way as PFG1, PFG2 makes use of ALC measurements together with visibility and relative humidity from a meteorological station. The current version of PFG2 retrieves fog alerts at a vertical and temporal resolution of 15 m and 1 min, respectively. PFG2 has also been entirely upgraded to Python 3 and the main advances (compared to PFG1) are:

i) A more efficient memory management.

ii) The development of the specific radiation (PFG2-RAD) and lowering stratus (PFG2-STL) fog modules.

iii) An assessment of the fog life cycle (discriminating between formation and mature stages).

iv) New output visualization options (operational and reanalysis mode). Note that we only present PFG2 outputs visualizations in reanalysis mode in this study. The operational mode corresponds to a simplified version with the visibility,



attenuated backscatter profiles between 0 and 400m, alert levels and fog type retrieved from PFG2 together with the status of the algorithm.

The methodology of the PFG2 algorithm (Figure 2) is divided into three main steps:

a) PFG2 is "turned ON" when the relative humidity measured at ground level exceeds a value of 85 % for a period of at least 10 min.

b) The visibility allows to discriminate between the formation and mature fog stages. If the visibility is greater than 1000 m for a period of at least 10 min, a fog
formation module is activated.

c) The distinction between RAD and STL fog type during the formation stage is based on the cloud fraction analysis deduced from the ALC measurements. If the two-hour averaged cloud fraction between 0 and 1000 m a.g.l. is greater (lower) than 50 %, the STL (RAD) formation calculation is activated. To reliably
distinguish between RAD or STL fog situation, the cloud fraction calculation is updated every hour.

### b) Radiation fog module

PFG1 was initially designed to monitor the early stages of RAD fog events by analysing
the rate of change of the aerosol-particle hygroscopic growth derived from ALC measurements in near real time. An in-depth analysis of the performance of PFG1 (see Section 5a for details about the methodology and Figure 9) gave useful insights. Overall, PFG1 performance at SIRTA over 128 RAD fog events between 2011 and 2019 had a hit rate of 70 %. At the Paris-Roissy, Vienna and Zurich airport sites, hit rates of about 90 %
were achieved, while the performance was markedly lower than 50 % at the airport of Munich (37%) and Zurich (31%). This analysis reveals that a substantial number of RAD fog events were not detected at SIRTA and the Munich and Zurich airports using the PFG1 algorithm. Figure 3 shows histograms of visibility at both 4 and 20 m, together with attenuated backscatter profile statistics for the first 60 minutes of all RAD fog events that
occurred at SIRTA during the observation period. Although observed values of horizontal



visibility can fluctuate at the scale of minute, the visibility measured at 4 m is lower than 1000 m for hit (missed) RAD fog events for ~ 90% (~ 75%) of the time. At 20 m, this situation is different since it represents 80% (25%) for hit (missed) RAD fog events. This discrepancy can be explained by the fact that RAD missed events correspond to shallow

radiation fog layers, while RAD hit events are associated with thick radiation fog layers. The distinct distributions of the attenuated backscatter profiles related to hit/missed RAD fog events, respectively confirm this conclusion (Figure 3 c-d). Attenuated backscatter profiles associated with missed RAD events are on the order of $1 \times 10^{-6}$ $sr^{-1}.m^{-1}$, except for the first range gate near the surface with values around $1 \times 10^{-5}$ $sr^{-1}.m^{-1}$ (Figure 3d).

In contrast, the attenuated backscatter profiles associated with the thick RAD fog layers that resulted in PFG1 "hits" show high values ranging from $1 \times 10^{-3}$ $sr^{-1}. m^{-1}$ at the surface to $1 \times 10^{-6}$ $sr^{-1}.m^{-1}$ at 100m, and are around $1 \times 10^{-7}$ $sr^{-1}.m^{-1}$ for higher altitudes (Figure 3c). These two regimes are typical of thin and thick RAD fog events, respectively (Haeffelin et al., 2016). The thin RAD fog events occurred near the surface in a very

shallow hydrated layer, where the ALC measurements are not able to monitor the aerosol hydration. According to Kotthaus et al. (2016), observation in the first range gate of the Vaisala CL31 measurement is of poor quality due to incomplete optical overlap. Therefore, PFG1 has difficulties to provide alerts for shallow radiation fog layers. Thin RAD fog events occur frequently at Munich and Zurich airports.

To incorporate also very shallow fog layers, a new approach based on a fuzzy logic algorithm has been implemented in PFG2. Here, the fuzzy-logic algorithm (Mendel, 1995) transforms non-linear data into scalar outputs referring to low, moderate and high fog alerts (hereafter referred to as LOW, MOD, HIGH, respectively). The fuzzy-logic algorithm has been selected due to its simple implementation and its low computational

cost. Here it relies on a combination of visibility measurements and attenuated backscatter ratio gradient (*RG* in Haeffelin et al., 2016). RG allows to monitor the aerosol activation process and is derived from a reference ALC-attenuated backscatter profile determined during pre-condition-fog conditions. For each considered alert, a typical range of values is assigned to the visibility and *RG* variables. Each range of values is expressed as a

membership function (MBF) and finally combined in a process called aggregation (A). The fuzzy logic method employed in the PFG2-RAD module uses one dimensional trapezoidal MBFs (*F*) to calculate the aggregation score that describes how well the





observations characterize the imminent fog formation. The general expression of the aggregation score $A^{level}(t)$ as a function of MBFs and alerts is given in equation 1:

$$A^{level}(t) = \frac{1}{2}\left(F_{visi}^{level}(t) + F_{RG}^{level}(t)\right)$$

(equation 1)

where *level* refers to the considered alert and $F_{visi}$ (resp. $F_{RG}$) represents the MBF associated with the visibility (resp. RG).

The MBFs are assumed to have trapezoidal shape and are described as follows:

$$Trap(x, x_1, x_2, x_3, x_4) = \begin{cases} 0, & (x < x_1)\ or\ (x > x_4) \\ \dfrac{(x - x_1)}{(x_2 - x_1)}, & x_1 < x \leq x_2 \\ \dfrac{(x_3 - x)}{(x_4 - x_3)}, & x_3 < x \leq x_4 \\ 1, & x_2 < x \leq x_3 \end{cases}$$

(equation 2)

where $x$ is the considered variable, $x_1$ and $x_4$ (resp. $x_2$ and $x_3$) the lower (resp. upper) corners of the trapezoid. Note that the complete parameters associated with the trapezoidal functions for fog formation in PFG2 are given in Table 2. The final score
$A^{level}(t)$ in equation 1 is converted to a RAD fog alert level (LOW, MOD, or HIGH) by assigning the alert with the maximum score (i.e. maximum rule value).

In complement to the PFG2-RAD module, the analysis of RG values together with its thickness allows to discriminate if a thin or thick fog layer will occur. A thick RAD fog layer is characterized by RG values greater than $4e^{-4}$ sr$^{-1}$.m$^{-1}$ over a layer of at least 30 m
thickness. If these conditions are not met, then it is considered as a thin RAD fog layer situation. These values were empirically derived from about ten RAD fog events analysed at SIRTA. Discrimination between thin or thick RAD layers is only performed for RAD HIGH alerts.





**c) Stratus lowering module**

The PFG2-STL module is again based on a fuzzy logic algorithm. Since low stratus clouds can be close to the ground for hours before their cloud base height starts to descend whereby causing a fog event, exploiting the attenuated backscatter ratio gradient as for PFG2-RAD does not provide useful insights. Hence, the PFG2-STL fuzzy logic algorithm

relies on a combination of visibility and cloud base height (CBH) observations (Figure 4). The PFG2-STL module uses one dimensional trapezoidal MBFs ($F$) together with weights ($w$) to calculate the aggregation score as described in equation 2:

$$\begin{cases} A^{level}(t) = \dfrac{w_{visi}(t).F_{visi}^{level}(t) + w_{CBH}(t).F_{CBH}^{level}(t)}{w_{visi}(t) + w_{CBH}(t)}, & during\ CBH\ lowering \\[3mm] A^{level}(t) = \dfrac{1}{2}\big(F_{visi}^{level}(t) + F_{CBH}^{level}(t)\big), & during\ CBH\ lifting \end{cases}$$

(equation 3)

where *level* refers to the respective alert, $F_{visi}$ (resp. $F_{CBH}$) represents the MBF associated with visibility (resp. CBH) and $w_{visi}$ ($w_{CBH}$) is related to the weight given to the visibility (resp. CBH).

The weights are determined empirically from the temporal gradient of both visibility and CBH variables, considering the 60 min prior to STL fog events that occurred at SIRTA

between 2011 and 2019. The weights are standardized with a linear scaling as follows:

$$w = \frac{x[i] - x_{min}[i]}{x_{max}[i] - x_{min}[i]}$$

(equation 4)

, where $x$ is the original value of the considered variable (temporal evolution of visibility or CBH), and $x_{min}$ ($x_{max}$) is the minimum (maximum) bound of $x$. The boundaries

employed in this study are 0 to -2500 m.h$^{-1}$ for the visibility gradient, and 0 to -50 m.h$^{-1}$ for the CBH gradient, respectively (Figure 4 a-b)). Note that these thresholds may need to be adapted for sites with very different fog characteristics. The final score $A^{level}(t)$ in equation 3 is converted to a STL fog alert level (LOW, MOD, or HIGH) by assigning the alert with the highest score (i.e. maximum rule value).



## 4 – Case studies


Here we present the ability of the PFG2 algorithm to anticipate the alert level for three different meteorological situations prior to fog formation at Munich airport, SIRTA observatory, and Zurich airport.

Figure 5 shows the time series measurements and the corresponding alert level outputs
from PFG2 during a thin radiation fog formation on March 3$^{rd}$ 2015 at Munich airport. Weather conditions are representative of a RAD fog event, with a decrease (increase) in temperature (relative humidity) at the surface level in response to a radiative cooling during this late winter afternoon. The reference attenuated backscatter profile is at the order of 1 x 10$^{-7}$ sr$^{-1}$. m$^{-1}$, with a reference RH$_{ref}$ = 71 %. The altitude of the maximum
gradient (*Hmax* in Haeffelin et al., 2016 – fuchsia dots in Figure 5-e) marks the atmospheric level with the most rapid hygroscopic growth of aerosol particles. The aerosol hydration (gray contour in Figure 5-e) represents the entire layer which is hydrating before the attenuated backscatter exceeds 1 x 10$^{-5}$ sr$^{-1}$. m$^{-1}$. In this case study, the first stages of the aerosol hydration occur above 300 m reaching layers near the surface
at about 45 min before the onset of the RAD fog event at 18:00 UTC. This thin fog layer would have been missed by PFG1 which is only based on the evolution of the rate of change of the aerosol-particle hygroscopic growth and only shown a few low to moderate alerts before 18:00 UTC. The combination of the visibility and the attenuated backscatter ratio gradient in the PFG2-RAD fuzzy logic algorithm, however, clearly improves the
anticipation of such very shallow fog layers. With PFG2, the first low alerts occur 2h before the fog onset, while the high alerts appear about 15 minutes ahead of the fog event (Figure 5f). Further the automatic assessment of the fog layer thickness classifies the event as a thin fog layer (Figure 5g).

Figure 6 is the same as Figure 5 for the case of 31 October 2015 at SIRTA. This is a
classic thick RAD fog event. During the night of 30$^{th}$ to 31$^{st}$ October, radiative cooling at the surface occurred during strong anticyclonic conditions located over the Paris region. As shown in Figure 6, the temperature (relative humidity) decreased (increased) from ~13 °C to 8 °C (from ~80 to 99 %), in presence of low-wind (< 4 m.s$^{-1}$) and clear sky conditions during the first part of the night. As a result of the radiative cooling, the
visibility was reduced, leading to a fog onset at 01:40 UTC. Overall, the PFG2-RAD





module performs well for this fog event since it has gradually delivered low to high alerts about 6 hours before the fog onset. The reference attenuated backscatter profile is on the order of 1 x $10^{-7}$ $sr^{-1}.m^{-1}$, with a reference $RH_{ref}$ of 68 %. The aerosol activation started in altitudes ranging from 50 to 200 m a.g.l. For this RAD fog case, the first HIGH alert

occurred ~100 min before fog onset, and the PFG2-RAD algorithm correctly identified the fog type as a thick radiation fog layer.

Figure 7 shows a classic STL fog event with a cloud base that fluctuates around 100 m a.g.l. four hours before reaching the surface, causing a fog onset at 19:00 UTC on 12 November 2015 at Zurich international airport. This STL fog event is characterized by

very high RH (> 95 %) at the surface (Figure 7-a) and a 100 % cloud fraction between 0 and 1000 m a.g.l. over the two hours prior to the event (Figure 7-c-d-e). This STL fog event is characterized by a visibility which decreases from 4 x $10^3$ m to 2 x $10^3$ m over the course of more than 3 hours (15h00 – 18h30 UTC) before dropping rapidly (18:30 – 19:00 UTC) when the cloud base reached the surface (Figure 7-e). For such an STL fog

formation, PFG2 alerts increase gradually from LOW to MOD (15h00 – 15h41 UTC), and then from MOD to HIGH (15h42 – 18h32 UTC), with HIGH alert reported more than 25 min prior to the fog onset.

**5 – Quantitative assessment of PFG2 performance**

The performance of the PFG2 algorithm is evaluated at 5 European sites, namely SIRTA, and the airports at Vienna, Munich, Zurich, and Paris-Roissy.

**a) Assessment methodology**

A specific assessment methodology has been designed to evaluate alerts provided by the PFG2 algorithm. The corresponding diagram of the PFG2 assessment methodology is

shown in Figure 7, while the overall assessment framework is described hereafter:

1. Since visibility measurements may fluctuate rapidly, only fog events retrieved by the Tardif and Rasmussen (2007) algorithm are evaluated.





2.  There are two possibilities to define the total assessment period. Either it ends with a previous fog event, or by reaching a set time limit of 3 (24) hours to agree with the time involved in RAD (STL) fog formation physical processes (Dupont et al., 2012; Haeffelin et al., 2016). All other alerts occurring outside this period are not considered.

3.  Although PFG2 is able to deliver an alert every minute, PFG2 performance is assessed on sub-periods of 45 minutes. The rationale for this approach is that PFG2 allows to track in near-real time physical key parameters at play within fog formation processes, which evolve over longer periods of time (longer than one minute). During a 45 min sub-period, the PFG2 alerts are transformed into an alarm (LOW, MOD, HIGH) that aims to summarize the global behaviour of alerts. Alarms are defined as follows:

    o   A minimum of 10 alerts (N) in a 45 min period is required to trigger an alarm.

    o   If one or more alert levels are present in a sub-period and exceed N, then the alert with the highest level defines the alarm reported.

    o   If the first 45 min sub-period before fog onset ends with a HIGH alert and the gradient of the alert level over the last 15 minutes is null or positive (e.g. LOW to MOD, and MOD to HIGH), then a HIGH alarm is assigned.

    o   After an episode of fog, a period of 90 minutes (two 45 min sub-periods) is removed from the performance analysis procedure if no new fog event occurs in that time. Given the visibility exceeds 1000 m during these 90 minutes after the event, PFG2 automatically calculates a formation risk (Figure 2). However, both ALC and meteorological measurements may present the same signatures as for "real" fog formation (e.g. during fog dissipation by lifting), and may mislead the performance analysis.

    o   The successive sub-periods presenting the same alarm levels are gathered in a single alarm (e.g. two consecutive HIGH alarms are counted as one).

4.  Finally, PFG2 performance is assessed based on the alarm retrieved. If a HIGH, (LOW, MOD or NONE) alarm is encountered before a fog event, it is considered as a hit (miss). If a HIGH alarm is encountered at another period with no consecutive HIGH alarm up to the start of a fog event, it is considered as a false





alarm. Note that the LOW and MOD alarms are not considered for the quantitative
       assessment of PFG2 performance. These alarms are intended as indicators of
       conditions favourable for fog formation, but without specific lead times.

**b) Application to SIRTA and European airport sites**

The quantitative assessment of PFG2 algorithm performances at the European sites is
       presented in Figure 9. It is based on a contingency table analysis and the two following
       categorical statistics:

$$Hit\ rate = \frac{Hits}{Hits + Misses}$$

and,

$$False\ Alarm\ ratio = \frac{False\ alarms}{False\ alarms + Hits}$$

.

       Overall, one can note from Figure 9-a that the PFG2-RAD performance is greatly
       improved compared to PFG1, regardless of the site. The new fuzzy logic algorithm allows
       to detect 100 % of RAD fog events at SIRTA and Paris-Roissy, Vienna, and Zurich
airports, and 93 % at the Munich airport. RAD events with a shallow fog layer are well
       anticipated by the PFG2-RAD module and allows to correctly detect the 30 % missing at
       SIRTA (up to 60 % at Munich and Zurich airports) in comparison to PFG1. False alarm
       ratios are on the order of 10%, with slightly higher values at SIRTA (14%) and the Vienna
       airport (19%).

Figure 9-b presents the PFG2-STL module assessment at the test sites. Again, the
       statistics are mostly similar between locations. While PFG2-STL does not miss any STL
       events at the airport sites, at SIRTA the hit rate is very slightly reduced to 96 %. This
       demonstrates the efficiency of the fuzzy logic algorithm integrated into PFG2-STL
       module. Overall, the false alarm rate associated with PFG2-STL module is 26 % for the
114 STL fog events at SIRTA between 2011 and 2019. The statistics are similar for Paris-





Roissy (26%), whereas it becomes 10 % at Munich, 40 % at Vienna, and 43 % at Zurich. However, these results must be strengthened with a more substantial database for the different airports which only present a few cases of STL over one or two fog seasons (Table 1).


### c) First high alerts characterization at SIRTA

Another important parameter of the statistical PFG2 assessment relies on the characterization of the first HIGH alert that results in a subsequent fog event during periods of hits. Here, the first high alert in the longest block of high alerts since the start
of a fog event is analysed over a 180 min period. Figure 10 shows the distribution of these first HIGH alerts for both RAD and STL fog events at SIRTA. Each hour is characterized by a different regime. The probability that a first "true" high alert occurs more than 2 hours before a fog event is relatively low, representing about 20% (5%) for RAD (STL) events. These probabilities are doubled between -120 and -60 min (RAD ~ 40 %; STL ~
10%), while it sharply increases (until to reach 100%) over the last hour prior to a RAD / STL fog event. Here, the discrepancies between the first HIGH alerts for the PFG2-RAD and -STL modules highlight the difference in terms of dynamics between the two fog types. Radiative fog events occur most of the time during night-time radiation cooling, characterized by low winds and high-pressure conditions. The hygroscopic growth of
condensation nuclei is progressive and allows PFG-RAD to anticipate well the related fog events by combining the visibility and the RG measurements. However, STL fog events may oscillate a few tens of meters above the surface before lowering and leading to a fog. This more "sudden" character is found in the first HIGH alerts of PFG2-STL which sometimes starts to retrieve them only a few minutes before the fog onset. As a result,
PFG2 has already 40 % (against 25 %) chance to have delivered the first HIGH alert for the RAD (STL) module one hour prior to a fog formation.



## 6 - Conclusions

A second version of PARAFOG (PFG2) has been developed to retrieve pre-fog alert levels and to discriminate between RAD and STL fog situation based on the analysis of ALC and meteorological station measurements in near real time. Two distinct modules have been developed to closely monitor the evolution of RAD and STL fog events which involve different physical processes. They rely on innovative fuzzy logic schemes that aim to combine both visibility and ALC measurements through one dimensional trapezoidal membership functions (and weights for the STL module) to characterize the fog formation threat level. In addition to these two main modules, some important advances have been carried out within PFG2 such as the redesign of the code in Python 3, a new memory management, together with new visualization outputs for both operational and research purposes.

Overall, the pre-fog alert levels retrieved by both the PFG2-RAD and -STL modules at SIRTA, and both Munich and Zurich airports are found to be consistent with the local weather analysis. Pre-fog alert level gradually rises from LOW to MOD, and then from MOD to HIGH as one gets closer to a fog event and the visibility decreases. The HIGH pre-fog alerts are found to occur between 30 and 60 minutes prior to fog formation regardless of the fog type considered, whereas the associated thin/thick discrimination matches well with RAD fog events.

An original approach to assess the performance of the pre-fog alert levels retrieved by both the PFG2-RAD and -STL algorithms has been subsequently proposed to support these results. This analysis is based upon comparisons of predicted and observed fog events over sub-periods of 45 minutes and the associated alarms deduced from the raw alert analysis. About 250 (up to 50) fog events that occurred at SIRTA (Munich, Zurich, Vienna and Paris-Roissy airports) between 2011 and 2019 (2014-2017) have been processed to assess the performance of the PFG2 algorithm. The retrieved pre-fog alert levels by the new PFG2-RAD module greatly improve the performance in comparison to PFG1 that failed to detect shallow fog events due to poorly defined ALC measurements in the first range gates. The PFG2-RAD module presents a hit rate of about 100 % and a false alarm ratio on the order of 10 % regardless the considered site. The retrieved pre-fog alert levels by the PFG2-STL algorithm are also defined by a hit rate of about 100 %





and a false alarm ratio on the order of 30 %. Finally, the first HIGH alerts that result in a subsequent fog event are found to occur for periods of time ranging from -120 minutes to fog onset, with first HIGH alerts occurring earlier for RAD than STL cases.

These encouraging results attest a good performance of the PFG2 algorithm which
warrants an extended application of the method at more locations. Implementing the PFG2 algorithm at the European scale, via the PROBE COST action (http://www.probe-cost.eu/) and the E-PROFILE network, will help whether the statistics obtained in this study are generally representative. In addition, it should be examined if and how other observations could improve the algorithm performance. For example, the combination of
a cloud radar and a microwave radiometer in near real time to retrieved the minimum amount of liquid water path which is necessary for a fog to remain at the surface (Toledo et al., 2021), could be implemented to estimate the fog dissipation probability and enhance PFG2.








**Data availability**

CL31 data, surface meteorological parameters and visibility measurements at SIRTA can be accessed from the SIRTA public data repository that is accessible online at http://www.sirta.fr. The data policy and a data download are available from the website.


**Author contribution**

JFR, MH, and JCD developed the concept of the paper. JFR designed the methodology, performed the analysis, and interpreted the results. MH, JCD, FT, and SK, contributed to
the design and discussion of the work. MAD and JFR performed the python 3 update and designed the main code structure of PFG2. JFR, with contribution from all authors, prepared the manuscript.

**Acknowledgements**

The contribution of the first author and the PFG2 project were supported by the "Direction Générale de l'Armement" under grant DGA 2018 60 0074. Felipe Toledo acknowledges the French Association Nationale Recherche Technologie (ANRT) and the company Meteomodem by their funding contribution. The authors would like to acknowledge Rafael Eigenmann and Ulrich Goersdorf (DWD) for kindly providing the data for Munich
airport, Maxime Hervo (Meteoswiss) for kindly providing the data for Zurich airport, Philipp Kneringer (Universität Innsbruck) for kindly providing the data for Vienna Airport, and Météo-France for kindly providing the data for Paris-Roissy Airport. Also, we would like to acknowledge the PROBE COST action for setting up an EU-wide discussion framework regarding applications and methods for fog nowcasting methods
based on remote sensing measurements.



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





**List of Tables**

**Table 1:** Main characteristics of the different sites and instruments used in this study.

**Table 2:** The trapezoid parameters for both RAD and STL modules and their associated variables used in PARAFOG-v2.

**List of Figures**

**Figure 1:** Map of the different instrumental sites used in this study.

**Figure 2:** Flow chart of PARAFOG-v2 algorithm. RAD stands for radiation fogs, whereas STL refers to stratus lowering fog events. RH stands for relative humidity and $CF_{0-1000m}$ refers to the cloud fraction between 0 and 1000m.

**Figure 3:** Characterization of the first 60 minutes of radiation fog events recorded at SIRTA according to the performance of PFG1. Visibility distribution at 20m (a) and 4m
(b), and boxplots of ALC-attenuated backscatter profiles associated to hits (c) and misses (d).

**Figure 4:** Fuzzy logic scheme used in PARAFOG-v2 algorithm for the STL module. Weights associated with a) visibility and b) CBH gradients. HIGH alert membership functions for c) the visibility and d) the CBH values.

**Figure 5:** Time series presenting measurements and the corresponding retrieved alerts level outputs from PARAFOG-v2 during a thin radiation fog formation on $3^{rd}$ January 2011 at Munich airport. a) Temperature and relative humidity, b) visibility at 4m, c) the Cloud Fraction (CF) between 0 and 1000m over the last 2 hours, d) ALC-attenuated backscatter between 400 and 6000m, e) ALC-attenuated backscatter between 0 and 400m
(color contours) together with the altitude of the maximum gradient (fuchsia points) and the aerosol hydration (gray contours), f) alert levels retrieved from PFG2, and h) fog type and PFG2 status.





**Figure 6:** Time series presenting measurements and the corresponding retrieved alerts level outputs from PARAFOG-v2 during a thick radiation fog formation on 30-31 October 2015 at SIRTA. a) Temperature and relative humidity, b) wind speed, c) visibility, d) the Cloud Fraction (CF) between 0 and 1000m over the last 2 hours, e) ALC-attenuated backscatter between 400 and 6000m, f) ALC-attenuated backscatter between

0 and 400m (color contours) together with the altitude of the maximum gradient (fuchsia points) and the aerosol hydration (gray contours), g) alert levels retrieved from PFG2, and h) fog type and PFG2 status.

**Figure 7:** Time series presenting measurements and the corresponding retrieved alerts level outputs from PARAFOG-v2 during stratus lowering fog formation on 12 November

2015 at Zurich airport. a) relative humidity, b) visibility, c) the Cloud Fraction (CF) between 0 and 1000m over the last 2 hours, d) ALC-attenuated backscatter between 400 and 6000m, e) ALC-attenuated backscatter between 0 and 400m (color contours) together with the altitude of the maximum gradient (fuchsia points) and the aerosol hydration (gray contours), f) alert levels retrieved from PFG2, and g) fog type and PFG2 status.

**Figure 8:** Diagram of PARAFOG-v2 assessment methodology. The alert colors represent PARAFOG v2 outputs with red for high alert, orange for moderate alert and yellow for low alert.

**Figure 9:** PARAFOG scores for a) radiation and b) stratus lowering fog events for the SIRTA and EU sites. The hatched bars correspond to the scores obtained with

PARAFOG-v1.

**Figure 10:** Cumulative distribution of the first RAD/STL HIGH alert that resulted in a subsequent fog event during the last 180 minutes.





| Name | Location | Period | Number of fog events | | Instruments | Time resolution |
|---|---|---|---|---|---|---|
| | | | RAD | STL | | |
| SIRTA | (48.713°N; 2.208°E; 160m a.s.l.) | 2010.10.01 – 2020.01.01 | 128 | 114 | Vaisala CL31 | 3s / 30s |
| | | | | | Degreane DF20/20+/320 | 1min |
| Paris Charles de Gaulle airport | (49.025°N; 2.567°E; 119m a.s.l.) | 2015.11.18 – 2017.07.02 | 16 | 17 | Vaisala CL31 | 30s |
| | | | | | Degreane DF320 | 1min |
| Zurich airport | (47.457°N; 8.559°E; 432m a.s.l.) | 2015.08.19 – 2016.07.26 | 16 | 8 | Vaisala CL31 (vertical res. 10m) | 15s |
| | | | | | Vaisala PWD22 | 1min |
| Munich airport | (48.354°N; 11.786°E; 453m a.s.l.) | 2015.10.07 – 2016.04.10 | 30 | 9 | Vaisala CL31 (vertical res. 10m, 910nm) | 15s |
| | | | | | Vaisala FS11 | 1min |
| Vienna airport | (48.11°N; 16.585°E, 183m a.s.l.) | 2014.01.01 – 2014.12.31 | 17 | 9 | Vaisala CL31 | 12s |
| | | | | | Vaisala FS11 | 5min |

**Table 1:** Main characteristics of the different sites and instruments used in this study.










| | | Alert | x1 | x2 | x3 | x4 |
|---|---|---|---|---|---|---|
| RAD | Visibility [m] | LOW | 2000 | 3500 | 8000 | 10000 |
| | | MOD | 999 | 2000 | 3500 | 5000 |
| | | HIGH | 999 | 1000 | 2000 | 4000 |
| | CBH [m] | LOW | 60 | 110 | 200 | 250 |
| | | MOD | 40 | 75 | 110 | 150 |
| | | HIGH | 60 | 110 | 200 | 250 |
| STL | Visibility [m] | LOW | 6000 | 8000 | 12000 | 15000 |
| | | MOD | 3000 | 4000 | 8000 | 10000 |
| | | HIGH | 999 | 1000 | 4000 | 5000 |
| | Ratio Gradient [sr⁻¹] | LOW | $5e^{-5}$ | $1e^{-4}$ | $4e^{-4}$ | $1e^{-3}$ |
| | | MOD | $1e^{-4}$ | $4e^{-4}$ | $1e^{-3}$ | $5e^{-3}$ |
| | | HIGH | $4e^{-4}$ | $1e^{-3}$ | 1 | 2 |

**Table 2:** The trapezoid parameters for both RAD and STL modules and their associated variables used in PARAFOG-v2.

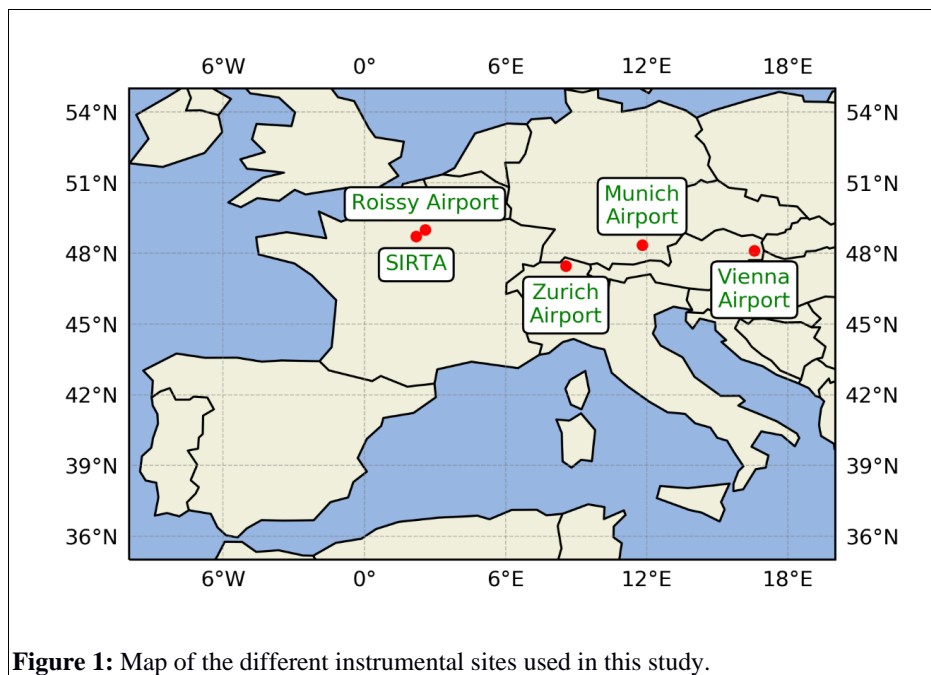

**Figure 1:** Map of the different instrumental sites used in this study.



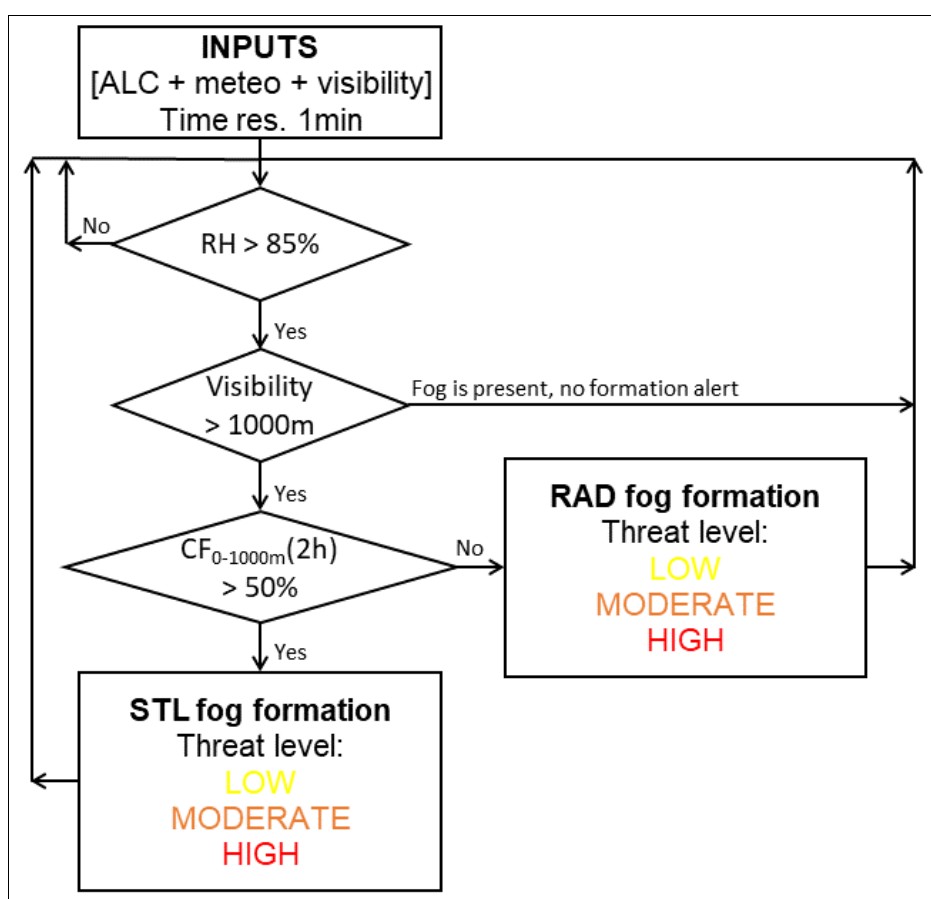

**Figure 2:** Flow chart of PARAFOG-v2 algorithm. RAD stands for radiation fogs, whereas STL refers to stratus lowering fog events. RH stands for relative humidity and $CF_{0-1000m}$ refers to the cloud fraction between 0 and 1000 m.



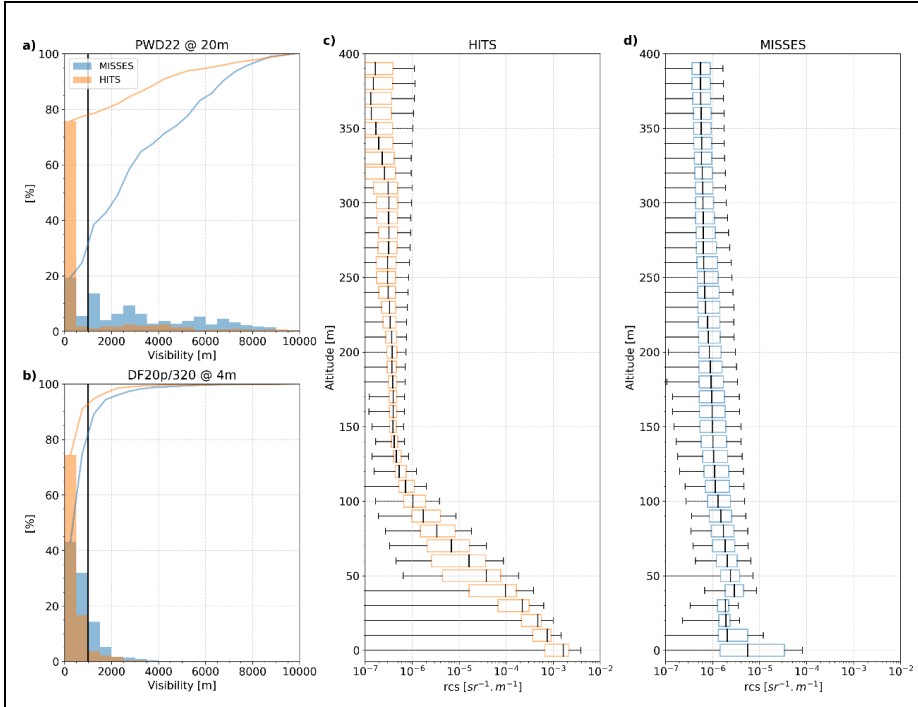

**Figure 3:** Characterization of the first 60 minutes of radiation fog events recorded at SIRTA according to the performance of PFG1. Visibility distribution at 20 m (a) and 4 m (b), and boxplots of ALC-attenuated backscatter profiles associated to hits (c) and misses (d).




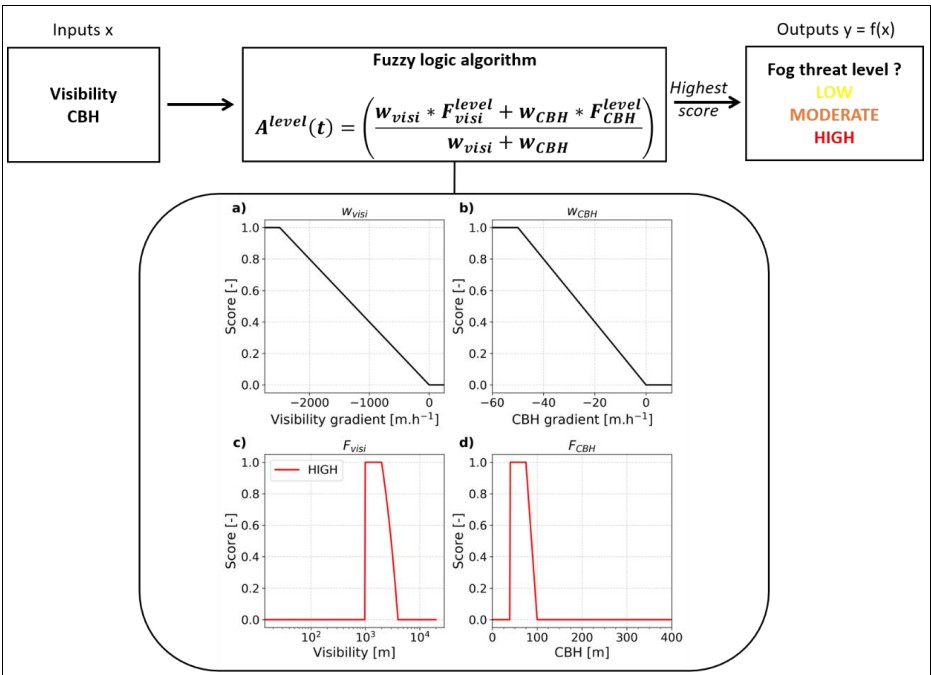

**Figure 4:** Fuzzy logic scheme used in PARAFOG-v2 algorithm for the STL module. Weights associated with a) visibility and b) CBH gradients. HIGH alert membership functions for c) the visibility and d) the CBH values.


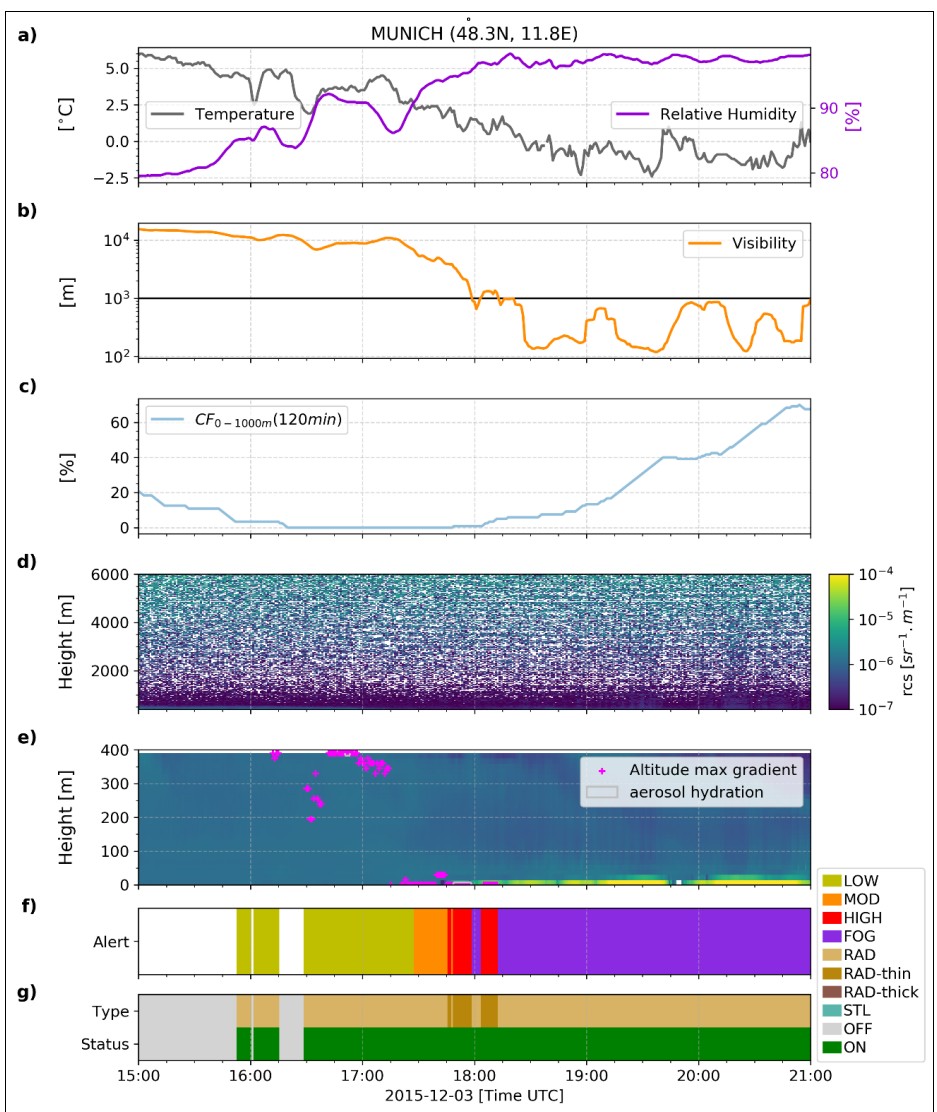

**Figure 5:** Time series presenting measurements and the corresponding retrieved alerts level outputs from PARAFOG-v2 during a thin radiation fog formation on 3rd January 2011 at Munich airport. a) Temperature and relative humidity, b) visibility at 4 m, c) the Cloud Fraction (CF) between 0 and 1000 m over the last 2 hours, d) ALC-attenuated backscatter between 400 and 6000 m, e) ALC-attenuated backscatter between 0 and 400m (color contours) together with the altitude of the maximum gradient (fuchsia points) and the aerosol hydration (gray contours), f) alert levels retrieved from PFG2, and h) fog type and PFG2 status.



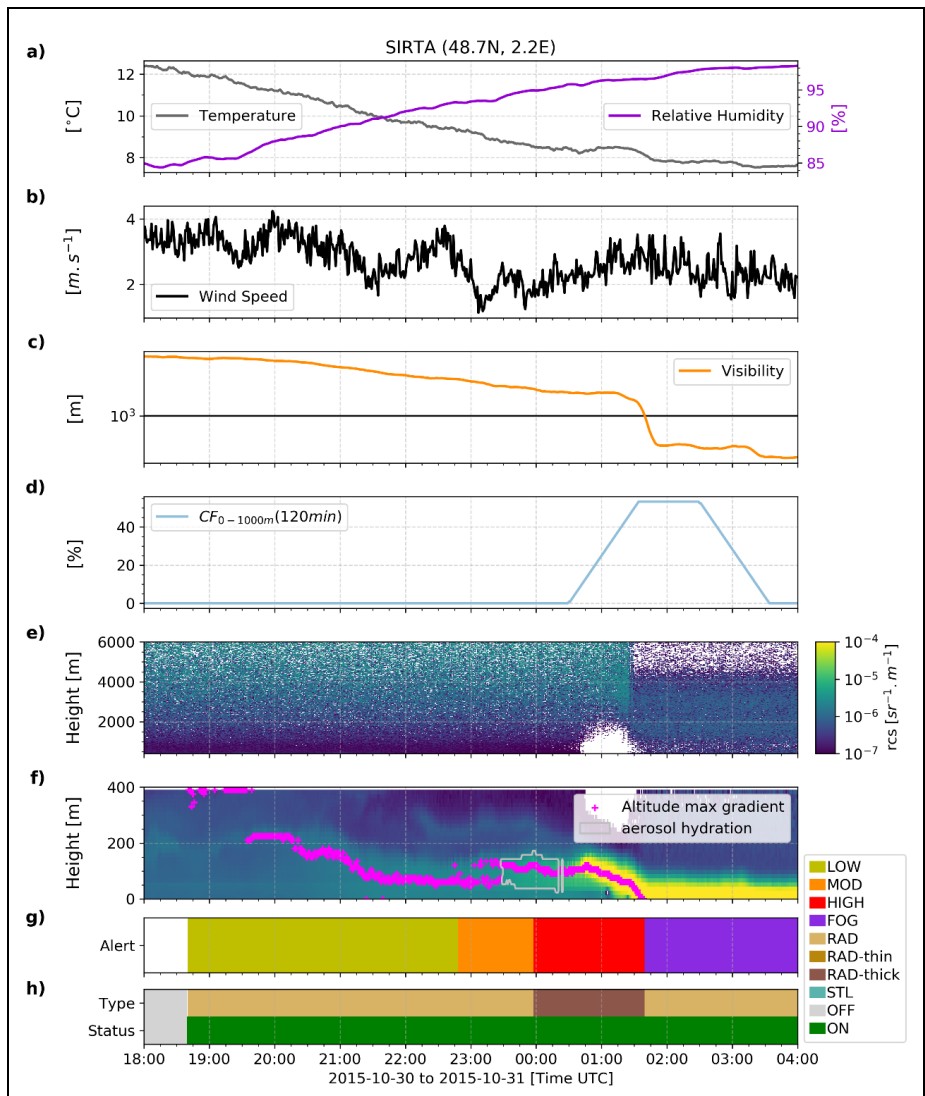

**Figure 6:** Time series presenting measurements and the corresponding retrieved alerts level outputs from PARAFOG-v2 during a thick radiation fog formation on 30-31 October 2015 at SIRTA. a) Temperature and relative humidity, b) wind speed, c) visibility, d) the Cloud Fraction (CF) between 0 and 1000 m over the last 2 hours, e) ALC-attenuated backscatter between 400 and 6000 m, f) ALC-attenuated backscatter between 0 and 400 m (color contours) together with the altitude of the maximum gradient (fuchsia points) and the aerosol hydration (gray contours), g) alert levels retrieved from PFG2, and h) fog type and PFG2 status.

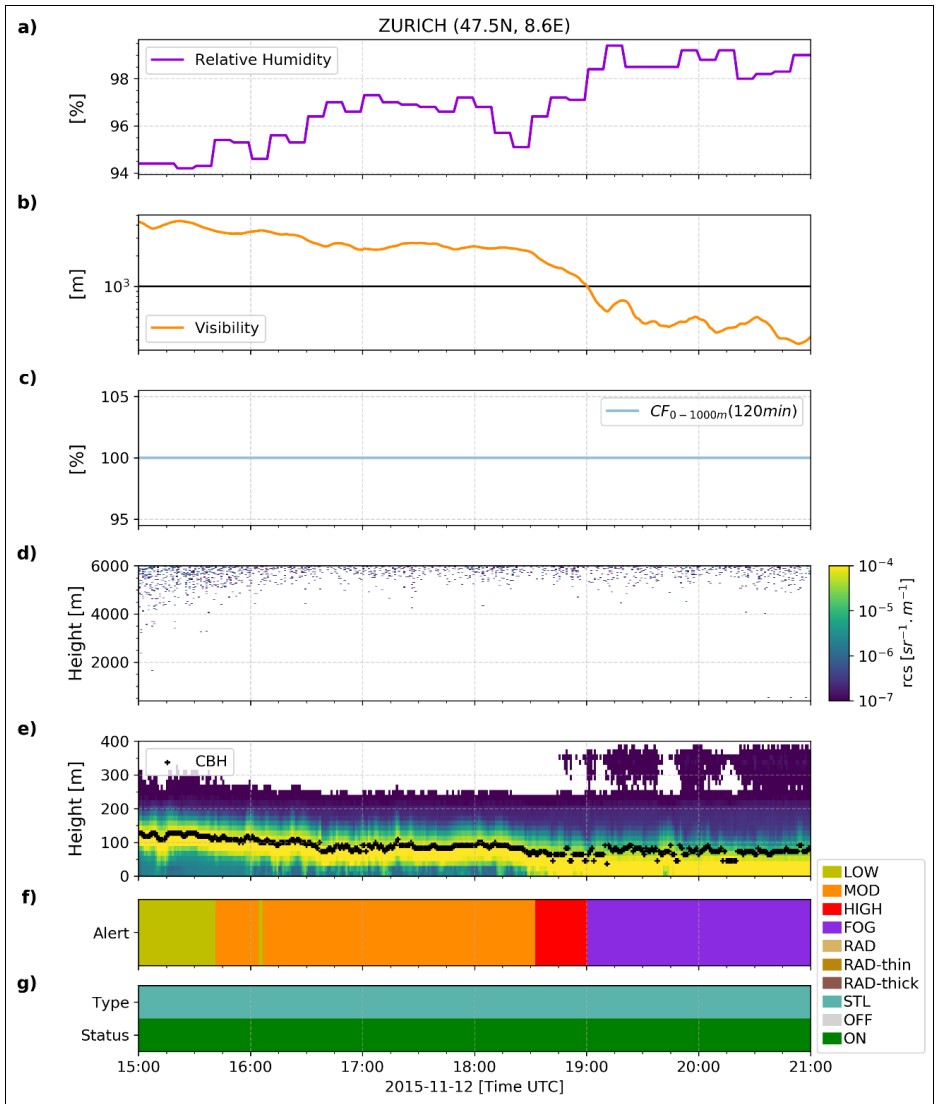

**Figure 7:** Time series presenting measurements and the corresponding retrieved alerts level outputs from PARAFOG-v2 during stratus lowering fog formation on 12 November 2015 at Zurich airport. a) relative humidity, b) visibility, c) the Cloud Fraction (CF) between 0 and 1000 m over the last 2 hours, d) ALC-attenuated backscatter between 400 and 6000 m, e) ALC-attenuated backscatter between 0 and 400 m (color contours) together with the altitude of the maximum gradient (fuchsia points) and the aerosol hydration (gray contours), f) alert levels retrieved from PFG2, and g) fog type and PFG2 status.


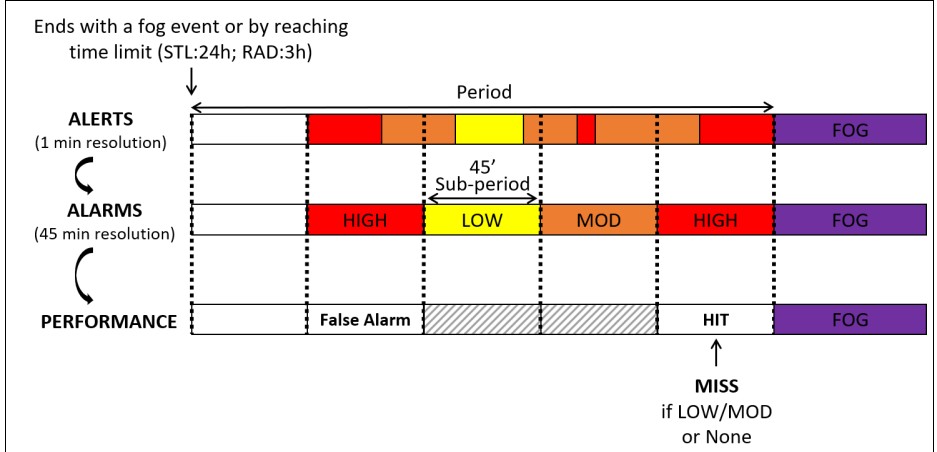

**Figure 8:** Diagram of PARAFOG-v2 assessment methodology. The alert colors represent PARAFOG v2 outputs with red for high alert, orange for moderate alert and yellow for low alert.



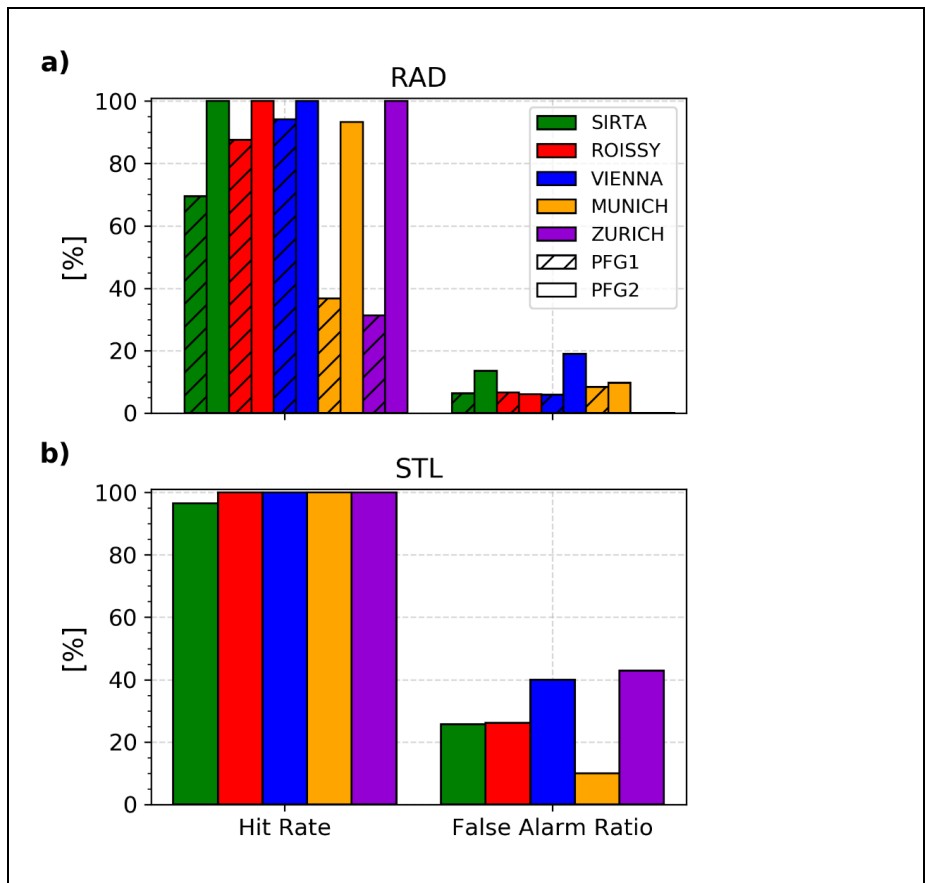

**Figure 9:** PARAFOG scores for a) radiation and b) stratus lowering fog events for the SIRTA and EU sites. The hatched bars correspond to the scores obtained with PARAFOG-v1.



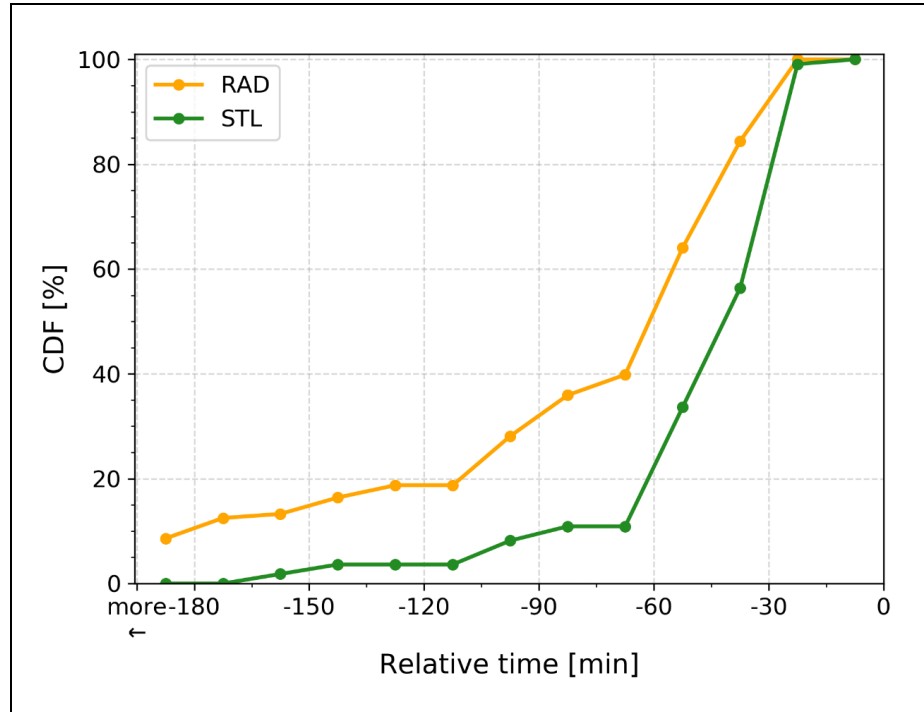

**Figure 10:** Cumulative distribution of the first RAD/STL HIGH alert that resulted in a subsequent fog event during the last 180 minutes.


