# Peer review of "PARAFOG v2.0: a near real-time decision tool to support nowcasting fog formation events at local scales"

_Atmospheric Measurement Techniques, 2021_

## Referee Comment (RC2)

**Review of article : « Parafog v2.0 : a near real-time decision tool to support nowcasting fog formation events at local scales » by Ribaud** *et al.**

**General comments:**

The article presents an evolution of a near real-time forecast system of fogs based solely on local observations made by sensors that are commonly available on airports (ceilometers, visibilimeters, surface weather stations). The original forecast system was published in 2016. The evolution proposed in the present article improves its performance by 1./ discriminating radiation and stratus-lowering fogs (using ceilometer data), and 2./ using a fuzzy-logic approaches for issuing alerts (low, medium or high probability of fog) in both cases.

The prediction of fogs on airports is a real issue. Fogs impair airport operations, reduce the capacity (number of flights allowed to land or take-off per unit time period), and generate costly delays and missed connections. Airport authorities would like to have reliable fog forecasts at ranges going from a few tens of minutes to about 12 to 24 hours in order to adapt the operations and mitigate the impact. Such forecasts are not yet available. The phenomenon is particularly difficult to predict by numerical weather prediction systems as the involve highly non-linear processes. Studies are ongoing in order to test the impact of improved model parametrisations (microphysics, turbulence, surface exchanges) and refined vertical/horizontal model resolutions (see for instance Philip et al., 2016). Ensemble-predictions with these improved models could in principle provide useful probability forecasts, but they will not become operational before years. Approaches based solely on local observations as in the present article are relevant for short term (minutes to a few hours) predictions. Signatures of the processes involved in fog formation under favorable fog conditions can indeed be detected. A main limitation for this kind of approach is that it does not give a precise time for the formation (a formation in an hour rather than in 15 minutes has a different impact on airport operations), but it nevertheless gives a highly valuable information.

The short-term alert system of fog formation proposed in the present article is thus of great interest as it could be easily implemented on current airports since it uses standard observation equipments already available. The performances reported in the article are very good. However, the performance scores (hit-rate and false-alarm ratio) used in the article confirm the warning system has been able to detect early for formation processes, but they do not measure the practical usefulness of the system in an operational context. As mentioned in the article, 10-minute alerts can vary quite a lot from one 10-minute time slot to the next. This variability is smoothed out by considering the prevalent alert in the last 45 minutes, which then becomes an alarm. The scores show the alarm achieves very good performances, but with a lead-time substantially reduced (up to 45 minutes between the first high probability alert and the issued high probability alarm). Scores measuring the relevance of the alerts or alarms for the prediction of fogs in 30 minutes, 1 hr, 1.5 hr, 2hrs... would be more relevant.

The article is rather well written, but its clarity could be greatly improved by giving or reminding definitions of variables before they are used (the definition of skill scores is given on page 15 but they are substantially mentioned before; the attenuated backscatter ratio gradient).

Overall, considering the importance of fog forecasts on airports, the inability of present systems to meet airport operators needs and the good scores of the PFG2, I consider the article deserves to be published. Small modifications would improve its clarity, and the addition, if possible, of skill-scores for 30-minute fog prediction ranges, would allow to better assess the particle usefulness of the system.

Minor comments

- Page 3, line 50: an international definition of fog by WMO exists (see <a href="https://cloudatlas.wmo.int/en/fog-compared-with-mist.html">https://cloudatlas.wmo.int/en/fog-compared-with-mist.html</a>) and should be cited here rather than AMS.
- Page 3, 1st paragraph: military operations could be added among fog impacts.
- Page 3, 2nd paragraph: the studies on improved version of operational NWP systems such as those reported by Philip at al., 2016, should be added at the end of the paragraph.
- Page 7, line 172: the article addresses RAD and STL fogs only. The authors claim they represent more than 90% of fogs on the study sites considered in the article. But they may represent much less on other sites (coastal sites for instance where advection fogs prevail). This should be stated here as it is a probable limit to the application of PFG2.
- Page 8, line 211: the definition or the meaning of hit-rate and false alarm should be given here as both notions are used in the following paragraph.
- Page 9, line 250: the RG is not defined. A reference is given, but a short summary of what is is would improve the readability of the article.
- Page 18: low-level wind speed could be added here as a relevant parameter that is measured at ground by weather stations or could be measured at low altitudes by small Doppler lidars.
- Page 27, Table 2. CBH parameters appear in RAD and Ratio Gradient in STL. There seams to be here a swap between lines. To be checked and corrected if swap confirmed.
- Page 35, Figure 8: the word FOG is unreadable in velvet cells.

**References**

Philip, A., Bergot, T., Bouteloup, Y., & Bouyssel, F. (2016). The Impact of Vertical Resolution on Fog Forecasting in the Kilometric-Scale Model AROME: A Case Study and Statistics, Weather and Forecasting, 31(5), 1655-1671. https://doi.org/10.1175/WAF-D-16-0074.1

---

## Author Comment (AC1)

**SCIENTIFIC COMMUNITY COMMENT: Philipp Körner**

**General comments:**

The article introduces the tool PARAFOG v2.0, which is used to predict the occurrence of fog at certain locations. It distinguishes between radiation fog and deep, lowered stratus clouds. The authors claim to be able to predict fog with a hit rate of almost 100%, with a false alarm ration of 10 and 30% respectively.

The methodology is a further development of PARAFOG, with a significantly better hit rate. Overall, the methodology is worth publishing. However, in my opinion, there are some major weaknesses, which I would like to discuss in more detail below. I will limit myself to the things worthy of criticism. Language, stile, title, credit to other authors etc. are well done. My main criticism relates to performance analysis. After reading the abstract, the reader is curious how the authors managed to achieve such a good hit rate. However, when reading the article, disillusion sets in.

**Specific Comments:**

- The weights of the fuzzy logic algorithm were derived from events at the SIRTA station. However, the same fog events were also used for performance analysis. There is no differentiation between calibration and validation period.

Fuzzy logic scheme such as pre-fog alert algorithm is part of IA but is different from machine learning technics. Here, we define explicit parameters based on our observations and analysis (i.e. more than 100 STL fog events).

The best validation is however the encouraging results we obtained at Paris-Roissy, Zurich, Munich and Vienna airports by using the SIRTA' weights (Figure 9b).

- Only the results for "high" fog alert are shown. What happens after a low or medium fog alert, is not mentioned. Does every low fog alert lead to a medium and this to a high fog alert? Are there very often false alarms for low or medium fog alerts? We do not know. Nevertheless, it would be interesting for the overall assessment.

We thank you for the suggestion. HIT/MISS definition are the parameters used to evaluate performance. Alerts are tools to provide an intuitive output for the user. Since only high alerts are defined based on optimization criteria, it is logical to only evaluate their performance. Medium/low alerts are not optimized, but they are included because they can indicate the user that conditions are not 100% risk free with respect to clear weather, in a qualitative way.

- In the abstract and also in the article itself, it is written about minute resolution. However, in the corresponding place it says that the minute resolution is used to make aggregations to 45 minutes. A fog forecast is therefore often only possible in the aftermath (reanalysis), as the authors also write themselves. The operational mode, is mentioned, but results are not shown. A prediction in a reanalysis, when both the event itself and events from the future have entered the algorithm, has only little value as a quality analysis.

We think there is a misunderstanding between alert and alarm concepts. Please refer to section 5a. Alerts are used for real time, while alarms are only used for performance assessment (Figure 8). There is no conflict between near real-time analysis and reanalysis.

What we evaluated is the quality of the parameters used to determine a HIGH alert using the reanalysis. It is true that this is not exactly the same as evaluating the predictive potential, but optimizing the variables based on this result should optimize the predictions. With this regard, Figure 9 shows the statistics of what a user should expect when observing HIGH alerts.

- The exact decision-making process for assigning an alert level to the 45-minute windows is described in the paper. All assessment steps described in 5.a) sound plausible for themselves. However, especially in 5.a)-3, it is not described how exactly the numbers are derived. Why do you choose exactly 45 minutes, exactly 10 alerts and form the gradient over exactly 15 minutes? It can be assumed that the values were determined on the basis of the fog events themselves. Is that the case? If so, on which ones? One can assume, that this further weakens the statement of the performance analysis.

The assessment methodology that we propose is entirely novel and specifically designed to the PFG2 tool. Like any method, it is however not perfect. We agree that the choice of some values used for assigning an alert level to the 45-minutes alarm period may be arbitrary. Here, 45 minutes provide a minimum time to react in case of eventual fog formation. Note that PFG2 performance farther than 45 minutes before fog formation decrease and this may be linked with fog nature and the method principles (use of observations only). Hence, we are satisfied to present a hit rate of about 100% in this last 45 minutes when the signal is clear. This also implies that observable in-situ formation signals almost always appear in the last 45 minutes, but may or may not appear before (at least with this instrumental setup). Regarding the use of 10 alerts or 15 minutes gradients, it is because these parameters are optimized to provide a good compromise between HIT/FA/MISS with the present evaluation scheme.

Despite all this, it constitutes today the first complete evaluation of PFG1 and PFG2 which represent an important achievement.

- line 382++ The removal of certain events from the performance analysis manipulates the same. If the model cannot handle the situations described, then this is a weakness of the model that can either be named as such or be improved. Filtering out these events afterwards is not a solution.

In reality we do not remove certain events. At this stage it is not possible to decide if fog will re-form or dissipate, based on the instrumental setup. To tackle this issue during this period, we may set alerts to NaN in the future version of PFG.

In summary: The method and also the performance analysis have numerous degrees of freedom. The number of fog events is very small. This potentially leads to an overfitting of the method. In addition, an unknown number of events were filtered out of the performance analysis. Don't get me wrong: the tool is probably good or even very good. It's just that the validation method used is not suitable to prove this quality. The paper may be published after major revision.

We thank you for the thorough comments and changes suggested in your review of our manuscript. We agree that the assessment methodology is specifically designed to the PFG2

tool. With this regard, statements about PFG2 performance have been rewritten in the revised manuscript to consider your overall remarks.

Please the revised text now reads in section 5.b:

"Note that this evaluation methodology has certain limitations. Arbitrary choices to consider only a 45 minutes alarm sub-period, or to have a minimum number of 10 alerts to trigger an alarm, may affect overall final performance. These parameters are optimized to provide a good compromise between hit/false alarm/miss with the present evaluation scheme, however, a sensitivity study may optimize the results. In addition, this method only evaluates the performance of PFG2 when fog events occur. Outside of these evaluation periods (3h for RAD and 24h for STL), PFG2 may deliver high alerts/alarms in pre-fog conditions such as during a stratus lowering having a cloud base height "stuck" a few tens of meters above the ground without leading necessarily to a subsequent fog event. This does not affect the PFG2 hit rate (number of hits or misses), but tend to underestimate the number of false alarms presented in this study."

**Minor remarks:**

- L143: how are 35 fog events a year fog- prone? Is that compared to other regions in France, or is there a general definition? In my ears, 35 fog events a year does not sound much.

**Corrected as suggested.**

- L366: "All other alerts occurring outside this period are not considered." Can you give an example? Erasing/not considering alerts in the aftermath is not suitable for a forecasting tool.

Please refer to the overall comment.

---

## Author Comment (AC2)

**Responses to Reviewer 1**

We thank you for the thorough comments and changes suggested in your review of our manuscript. Our point-to-point responses are developed hereafter, along with an indication of changes made in the revised version of the text.

**General Comments:**

The authors demonstrate an innovative technique of applying observed vertical gradient of backscatter profiles as well as cloud base height and visibility by way of a fuzzy-logic approach to predict the likelihood of fog onset. The new innovation serves to resolve a previously identified discrepancy in algorithm performance of shallow fog radiation fog identification. The methods are generally clear and well supported by the literature and through equations and figures. The authors provide several thorough, well-documented case study reviews demonstrating implementation followed by additional systematic statistical assessment at multiple stations. The approach appears to have applicability to additional stations globally where radiation and stratus-lowering fog are the predominant fog hazards, subject to some additional work needed to derive appropriate weights for the stratus lowering cases. The manuscript is very well written and the figures are of high quality.

I have listed several specific comments for the authors to address, but most concern suggestions to improve understanding of the methods and some clarifications. I recommend to accept following these minor revisions.

**Specific Comments:**

L64-65: Would suggest that fog decay/dissipation is also a major challenge for NWP. Implemented as suggested.

L67-68: Would suggest that the authors broaden this statement from "land-atmosphere" to "surface-atmosphere" interaction as similar dependencies of fog to surface turbulent flux occur over ocean as well as land surfaces. Would also suggest that the authors acknowledge the additional components that yield difficulty in fog prediction for NWP beyond surface-atmosphere interaction, vertical resolution and atmospheric boundary layer physics, namely: cloud microphysics parameterization, radiation parameterization and potentially shallow convection parameterization in coarser models. For limited area models, boundary conditions also help determine advection, another potentially significant contribution to fog presence/absence.

We agree. The revised text: "*The difficulties of NWP fog forecasting can be explained by the fact that fog events are driven by complex surface-atmosphere interactions in the atmospheric boundary layer, where vertical resolution of NWP models is still not high enough (e.g. Philip et al., 2016). Specifically, components that yield difficulties to fog prediction for coarser-grid models are cloud microphysics parameterization, radiation parameterization and potentially shallow convection parameterization, while for limited area models, boundary conditions also help determine advection, another potentially significant contribution to fog presence or absence.*"

Figure 2: This is a helpful diagram of the algorithm. I am slightly confused, however, why there is no sensitivity to wind speed or vertical shear in the algorithm? For example, there could be a condition of high relative humidity near the ground, but the speed of the ambient horizontal wind may generate sufficient turbulence to prevent fog formation? Have the authors evaluated this as a predictor?

*This is a relevant comment. Unfortunately, we did not investigate the sensitivity to low wind speed/shear in the PFG algorithm. We tried to keep PFG as simple as possible to make this tool widely and easily applicable. The comment has been, however, added in the perspectives: "For example, wind shear analysis could be used to support fog formation prediction by assessing the ambient horizontal wind speed and checking whether it may generate sufficient turbulence to prevent fog."*

L207-209: Using a two-hour averaged cloud fraction seems somewhat temporally coarse relative to the 1-min resolution of the algorithm (L188). Was this choice made because of the coarse frequency of the data source? Did the authors use METAR observations of cloud to make this two-hour average determination?

*The rationale for this approach is to keep PFG2 "stable" and avoid frequent changes from RAD to STL module (as it may mislead the user and may induce loss of continuity in alerts). Please note that we directly use the cloud base height retrieved by the ceilometers to calculate the cloud fraction.*

L219, L221: Did the Zurich airport have a hit rate of 90% or 31%? There seems to be a conflict here.

*Corrected! The revised text now reads: "At the Paris-Roissy and Vienna airport sites, hit rates of about 90 % were achieved, while the performance was markedly lower than 50 % at the airport of Munich (37%) and Zurich (31%)."*

Figure 3: I like how subplots (c) and (d) clearly indicate a clustering of patterns between the hit and miss incidents, complementing the explanation in the text well. In subplot (a), however, I do not understand how approximately 70 percent of all times observed during the verified fog periods for which PFG1 failed (misses) have observed visibilities greater than 1 km? On L168, the authors define events with visibility less than 1 km within at least three of five blocks of 10 min. Shouldn't there be no more than 40 percent of times showing visibilities exceeding 1 km? If this is not a mistake, then I think there needs to be much more clarification to the reader as to what is displayed in this figure. Also, it is a bit bewildering how nearly 20 percent of times during these 'missed' verified fog periods yield observed visibility exceeding 6 km. Given the time restrictions imposed by the authors on defining a fog event, how these kinds of outliers be explained physically? Are these outliers' artifacts of fog having decayed away completely but while time remains inside the 50 min window?

*We think there is a misunderstanding. Figure 3 subplot (a) shows the visibility distribution at 20m during the first 60 minutes of hit/missed radiation fog events recorded at SIRTA. This illustrates that most of the time (~ 70-75%) visibility at 20m associated to missed events is greater than 1000m (whereas it represents only ~25% at 4m in subplot (b)). Now if we combine this information with subplots (c-d), we can conclude that PFG1 missed events are related to shallow radiation fog layers as explained in the manuscript.*

Note that visibility measurements used for the Figure 3 a-b have a (native) time resolution of one minute. We did not use averaged blocks of 10min as in Tardif & Rasmussen (2007).

L277-278: This may be related my (mis) understanding of the alert level concept – but why is the RAD layer thickness discrimination only performed for RAD HIGH alerts? Is it not of interest at any likelihood (low, medium, high) to know whether the potential RAD fog event would be thick or thin? Or is it because the viability of the RG method breaks down under conditions of weaker likelihood?

In theory, it should be possible to discriminate between thick and thin events for moderate alert level as RG values should be greater than $4e^{-4}$ $sr^{-1}.m^{-1}$. However, the thick-thin discrimination was empirically derived from only few RAD fog events at SIRTA. It requires a more robust in-depth analysis before to be extended to moderate alerts. This could be applied in a near future within an updated version of PFG.

L288-289: I assume that the reason there are two separate Aggregation (A) equations to describe CBH lowering and lifting is that the former has some predictability through the time change of visibility and CBH quantities, whereas the latter does not (with respect to visibility and CBH alone)? Did the authors consider any other environmental predictor for CBH lifting? I think it would be helpful for the reader to have some understanding behind the authors' decision here regarding the two separate A equations. Regarding the weights, is the implication that future applications of this approach will require the assessment of a long period of fog climatology to generate these empirical values?

Weights were only defined for negative gradients in visibility and CBH. Stratus lifting will deliver no alert by using CBH lowering aggregation equation, even if the stratus is close to the surface. This makes it possible to avoid the discontinuity in the monitoring of stratus clouds and the resulting PFG2 alerts.

Regarding the weights (i.e. speeds), we assume there are area-dependant and it is better to have a substantial period of fog climatology to properly define them. However, the use of the default weights (i.e. from SIRTA) appears to work well at the different European sites in this study (Figure 9b).

To clarify the need for the two aggregation equations, the revised text now reads: "*As stratus clouds may oscillate a few tens of meters above the surface before lowering and leading to a fog, we define two aggregation equations to avoid any discontinuity in the alerts delivered.*"

L357: Section 5a describes important details about the methodology. I think this would be generally better suited earlier in the manuscript, before discussion of case studies. This might help with improving understanding of the 'alert' concept.

We forgot to mention the retrieved pre-fog alert levels in the overview of PFG2 in section 3.a. This should now improve the understanding of the alert concept. Note that we keep assessment methodology in section 5a.

The revised text now reads:

"*The methodology of the PFG2 algorithm (Figure 2) is divided into three main steps:*

*a) PFG2 is "turned ON" when the relative humidity measured at ground level exceeds a value of 85 % for a period of at least 10 min.*

*b) The visibility allows discriminating between the formation and mature fog stages. If the visibility is greater than 1000 m for a period of at least 10 min, a fog formation module is activated.*

*c) The distinction between RAD and STL fog type during the formation stage is based on the cloud fraction analysis deduced from ALC measurements. If the two-hour averaged cloud fraction between 0 and 1000 m a.g.l. is greater (lower) than 50 %, the STL (RAD) formation calculation is activated. To reliably distinguish between RAD or STL fog situation, the cloud fraction calculation is updated every hour.*

*d) PFG2 retrieves pre-fog alerts (low, moderate, high) every minute indicating the risk of fog formation."*

L375-390: It seems like the 'alarm' concept is really more of a decision to be made by an operational forecasting center given the 'alert' result of the algorithms. It's arguably beyond the scope of the scientific work presented here. It's OK to keep, but this could be one area to trim if adding content elsewhere, I think the quantification of "false alarms" is best done following the conventional contingency table methodology, as the authors do in the ensuing section.

We agree with the reviewer. The alarm concept is something we would recommend for final users, but this is tricky to implement in near real-time version for the moment.

Section 5c: Some lingering confusion here for me about the applicability of the alert in time. A HIGH alert multiple hours (or at least 45 minutes (L 369)) ahead of the first observation of fog is technically false alarm, yes? I think I understand the objective here to demonstrate that more first HIGH alerts happen nearer to the observed start of the fog, though Figure 10 doesn't seem to clearly distinguish the 'good' results from the 'bad'…Perhaps add some kind of marking at 45 minutes?

A hit happens whenever there is continuous high alerts at least 45 minutes before formation time, that are followed by a fog event. Here we try to find out the temporal distribution of the first HIGH alert that will result in a subsequent fog event during period of hits. With this regard, PFG2 alert occurrences and durations depend on pre-fog conditions. Sometimes fog formation may last several hours with high values of RG, or low CBH…leading PFG2 to deliver continuous HIGH alerts during a "long" period. If the HIGH alerts exceed 45min, it is possible to have multiple and successive HIGH alarms. According to the rule defined in section 5.a.3: "*successive sub-periods presenting the same alarm levels are gathered in a single alarm (e.g. two consecutive HIGH alarms are counted as one)*". So, it can be considered as a hit even if a HIGH alert appears one hour before the observed fog onset.

The main goal of such a tool like PARAFOG, is to anticipate as much as possible the occurrence of RAD/STL fog events. But also, to find the right compromise between delivering HIGH alerts only few minutes before an event and several hours before. Our HIT definition is based on operational parameters. Hence, if a fog event is correctly forecasted, even if it was earlier than 45 min before fog formation, we declare it as a HIT because it helped the user to prepare for the eventual fog case. This capability is the most important for air traffic controllers. A false

alarm, based on our definition, would happen when a fog event is predicted (HIGH ALARM) without subsequent formation of fog. This is what we use in our evaluation scheme (Figure 8). In this second version of PFG, we tried to mitigate the negative impact of a too long period with HIGH alerts, as it would probably mislead weather forecasters and air traffic controllers in their decision making. Today it seems to us that PFG2 is rather correctly optimized.

**Technical Corrections:**

L51: add a space between '1' and 'km', same issue on L138

Corrected.

L56: again: word choice; perhaps the authors mean 'also' or "as well"?

Corrected.

L75: Note that some LES models (e.g. Cloud Model 1 : https://www2.mmm.ucar.edu/people/bryan/cm1/) have the ability to incorporate cloud microphysical parameterizations.

Thanks for the reference.

L78: wording: allows (someone/thing) to monitor fog – OR – allows monitoring of fog... "allows" here needs to be followed by a noun; This grammar issue happens again on L93, L184, L203, L251 and elsewhere (I stopped seeking this out after L251).

Thanks for the grammar reminder! We checked and corrected all sentences containing "allow".

L92: recommend omitting 'true' – the measurements will also have some error

Corrected as suggested.

 L94: variable -> variables

Done.

L103-104: The phrase "PFG1 retrieves pre-fog alert levels" is a bit unclear at this point in the text, specifically the "alert levels". There is brief mention in the abstract describing the levels as low, medium and high. Do pre-fog alert levels refer to designation of a relative likelihood of fog based on observed pre-fog conditions that portend a certain type (RAD, STL)? Is it meant to describe likelihood within the next 15 minutes period? I would recommend to the authors to clarify this for the reader early on in the text. I did not get a clear context for the concept of the "alert level" early on and it affected my comprehension of the text and results later.

The revised text now reads: "*With this regard, PFG1 is able to retrieve pre-fog alert levels (low, moderate, or high alert) with a vertical resolution of about 15 m ranging from 0 to 400 m a.g.l. and time resolution of one minute.*"

L163: Please spell out '9' (any number less than 10). Same on line L168, potentially elsewhere where a single digit is used that is not a measured quantity.

Corrected.

L184: I think maybe the authors mean "key physical parameters"?

Corrected as suggested.

Section 3b: I think a full contingency table (hit, miss, false alarm, correct reject) would be useful for clarification and transparency here. The authors indicate hits and misses in the text here, but also important for more complete algorithm performance assessment are the complementary statistics of false alarms and correct rejections. The authors could choose to coordinate this with Figure 9, simply by expanding analysis to include FA and CR here and the M and CR in Figure 9 (i.e., show all four statistics for all four stations for PFG1 here, then make Figure 9 a complementary figure of comparison).

Categorical statistics such as hit rate or false alarm ratio scores in the manuscript are computed in order to highlight the PFG2 performance and represent the best arguments to promote the use of PFG2. It represents what a user should expect when observing HIGH alerts. We believe that indicating the full contingency table would not add much to the study since both hit rate and false alarm ratio are already based on hits, misses and false alarms. Note that the correct rejections were deliberately not taken into account since most of the time there are no fog events (that will result in high number of CRs).

To complete the performance assessment results we added the "full" contingency table with hit/miss/false alarm scores (Table 3 in the revised manuscript).

L274: Apparent errant period between the sr$^{-1}$ and m$^{-1}$; this also occurs elsewhere in the text.

Corrected.

Figure 5: Could you please shift the colorbar from subplot (d) downward to straddle between (d) and 9e) so that the reader knows to associate the plotted field in (e) with that backscatter colorbar? Otherwise it's not immediately obvious what is being plotted in subplot (e).

Corrected as suggested.

L343-351: Please follow a consistent format on the display of time (e.g. HH:MM UTC preferred over HHhMM UTC)

Fixed!

L403, 405: These definitions should appear earlier in the text, when the authors first introduce the contingency table terms.

Same remark as for reviewer 2. These definitions now are in section 3a.

Figure 9: Building on my earlier comment in Section 3b, this figure should illustrate all four standard contingency table statistics (H, M, FA, CR) to provide a well-rounded assessment of the results.

Please see earlier response in Section 3b.

---

## Author Comment (AC3)

**Responses to Reviewer 2**

We thank you for the thorough comments and changes suggested in your review of our manuscript. Our point-to-point responses are developed hereafter, along with an indication of changes made in the revised version of the text.

**General comments:**

The article presents an evolution of a near real-time forecast system of fogs based solely on local observations made by sensors that are commonly available on airports (ceilometers, visibilimeters, surface weather stations). The original forecast system was published in 2016. The evolution proposed in the present article improves its performance by 1./ discriminating radiation and stratus-lowering fogs (using ceilometer data), and 2./ using a fuzzy-logic approaches for issuing alerts (low, medium or high probability of fog) in both cases. The prediction of fogs on airports is a real issue. Fogs impair airport operations, reduce the capacity (number of flights allowed to land or take-off per unit time period), and generate costly delays and missed connections. Airport authorities would like to have reliable fog forecasts at ranges going from a few tens of minutes to about 12 to 24 hours in order to adapt the operations and mitigate the impact. Such forecasts are not yet available. The phenomenon is particularly difficult to predict by numerical weather prediction systems as the involve highly non-linear processes. Studies are on-going in order to test the impact of improved model parametrisations (microphysics, turbulence, surface exchanges) and refined vertical/horizontal model resolutions (see for instance Philip et al., 2016). Ensemble-predictions with these improved models could in principle provide useful probability forecasts, but they will not become operational before years. Approaches based solely on local observations as in the present article are relevant for short term (minutes to a few hours) predictions. Signatures of the processes involved in fog formation under favourable fog conditions can indeed be detected. A main limitation for this kind of approach is that it does not give a precise time for the formation (a formation in an hour rather than in 15 minutes has a different impact on airport operations), but it nevertheless gives a highly valuable information.

The short-term alert system of fog formation proposed in the present article is thus of great interest as it could be easily implemented on current airports since it uses standard observation equipment's already available. The performances reported in the article are very good. However, the performance scores (hit-rate and false-alarm ratio) used in the article confirm the warning system has been able to detect early for formation processes, but they do not measure the practical usefulness of the system in an operational context. As mentioned in the article, 10-minute alerts can vary quite a lot from one 10-minute time slot to the next. This variability is smoothed out by considering the prevalent alert in the last 45 minutes, which then becomes an alarm. The scores show the alarm achieves very good performances, but with a lead-time substantially reduced (up to 45 minutes between the first high probability alert and the issued high probability alarm). Scores measuring the relevance of the alerts or alarms for the prediction of fogs in 30 minutes, 1 hr, 1.5 hr, 2hrs... would be more relevant.

The article is rather well written, but its clarity could be greatly improved by giving or reminding definitions of variables before they are used (the definition of skill scores is given on page 15 but they are substantially mentioned before; the attenuated backscatter ratio gradient). Overall, considering the importance of fog forecasts on airports, the inability of present systems to meet airport operators needs and the good scores of the PFG2, I consider the article deserves to be published. Small modifications would improve its clarity, and the

addition, if possible, of skill-scores for 30-minute fog prediction ranges, would allow to better assess the particle usefulness of the system.

We thank the reviewer for this encouraging and rather positive general comment.

**Minor comments:**

• Page 3, line 50: an international definition of fog by WMO exists (see https://cloudatlas.wmo.int/en/fog-compared-with-mist.html) and should be cited here rather than AMS.

Corrected as suggested.

• Page 3, 1st paragraph: military operations could be added among fog impacts.

Corrected as suggested.

• Page 3, 2nd paragraph: the studies on improved version of operational NWP systems such as those reported by Philip at al., 2016, should be added at the end of the paragraph.

Thank you for the reference. Implemented as suggested.

• Page 7, line 172: the article addresses RAD and STL fogs only. The authors claim they represent more than 90% of fogs on the study sites considered in the article. But they may represent much less on other sites (coastal sites for instance where advection fogs prevail). This should be stated here as it is a probable limit to the application of PFG2.

Corrected as suggested. The revised text now reads: "*Note that it may represent much less on other locations such as for instance coastal sites, where advection fog prevails and where PFG2 is not designed to monitor them.*"

• Page 8, line 211: the definition or the meaning of hit-rate and false alarm should be given here as both notions are used in the following paragraph.

Definitions have been added as suggested.

• Page 9, line 250: the RG is not defined. A reference is given, but a short summary of what is would improve the readability of the article.

The revised text now reads: "*Here it relies on a combination of visibility measurements and attenuated backscatter ratio gradient (RG in Haeffelin et al., 2016). RG allows the aerosol activation process (i.e. proxy to monitor for hygroscopic growth dynamics) to be monitored and is derived from a reference ALC-attenuated backscatter profile determined during pre-fog conditions.*"

• Page 18: low-level wind speed could be added here as a relevant parameter that is measured at ground by weather stations or could be measured at low altitudes by small Doppler lidars.

This is a very relevant comment and it has been added in the perspectives: *"For example, wind shear analysis could be used to support fog formation prediction by assessing the ambient horizontal wind speed and checking whether it may generate sufficient turbulence to prevent fog."*

• Page 27, Table 2. CBH parameters appear in RAD and Ratio Gradient in STL. There seems to be here a swap between lines. To be checked and corrected if swap confirmed. Swap confirmed. Fixed.

• Page 35, Figure 8: the word FOG is unreadable in velvet cells. We tried to improve readability.

**References**

Philip, A., Bergot, T., Bouteloup, Y., & Bouyssel, F. (2016). The Impact of Vertical Resolution on Fog Forecasting in the Kilometric-Scale Model AROME: A Case Study and Statistics, Weather and Forecasting, 31(5), 1655-1671. https://doi.org/10.1175/WAF-D-16-0074.1

---

## Referee Report (RR1)

Review of revised version of manuscript « PARAFOG v2.0: a near real-time decision tool to support nowcasting fog formation events at local scales ».

This second version of the manuscript is much easier to read. There are still a few, minor, mistakes that should be corrected, but the article can be published once this is done.

Line 377 : the reference to Figure 7 is wrong. The reference should be Figure 8.

Table 2 on page 28 : The parameters for RAD (STL) are visibility and RG (visibility and CBH). RAD and STL in column 1 must be swapped.

Figure 5 on page 34 : the gray box is hardly visible.

---

## Author Response (AR2)

**Responses to Reviewer**

**We thank you for the thorough comments and changes suggested in your review of our manuscript. Our point-to-point responses are developed hereafter, along with an indication of changes made in the revised version of the text.**

**Comments to the author** :
Please take into consideration the suggested technical corrections by the reviewer before publication.

**Technical corrections :**
- Line 377 : the reference to Figure 7 is wrong. The reference should be Figure 8.

Corrected.

- Table 2 on page 28 : The parameters for RAD (STL) are visibility and RG (visibility and CBH). RAD and STL in column 1 must be swapped.

Swap confirmed. Fixed!

- Figure 5 on page 34 : the gray box is hardly visible.

We believe your are referring to Figure 5 panel b and aerosol hydration box. This is true. Unfortunately we cannot do better in this case of thin fog event and low hydration at ground level. If we increase the resolution of the box/line, the figure will become difficult to read during a thick fog event with a developed hydrated layer. We are aware of this, but it constitutes the « best » compromise today.